# Deletion of the *col*-26 Transcription Factor Gene and a Point Mutation in the *exo*-1 F-Box Protein Gene Confer Sorbose Resistance in *Neurospora crassa*

**DOI:** 10.3390/jof8111169

**Published:** 2022-11-06

**Authors:** Kenshi Hirai, Takuya Idemoto, Shiho Kato, Akihiko Ichiishi, Fumiyasu Fukumori, Makoto Fujimura

**Affiliations:** 1Department of Life Sciences, Toyo University, 1-1-1 Izumino, Itakura, Ora 374-0193, Gunma, Japan; 2Department of Food and Life Sciences, Toyo University, 1-1-1 Izumino, Itakura, Ora 374-0193, Gunma, Japan

**Keywords:** glucose repression, amylase, cellulose, glucose transporter, *Neurospora*

## Abstract

L-Sorbose induces hyperbranching of hyphae, which results in colonial growth in *Neurospora crassa.* The *sor*-4 gene, which encodes a glucose sensor that acts in carbon catabolite repression (CCR), has been identified as a sorbose resistance gene. In this study, we found that the deletion mutant of *col*-26, which encodes an AmyR-like transcription factor that acts in CCR, displayed sorbose resistance. In contrast, the deletion mutants of other CCR genes, such as a hexokinase (*hxk*-2), an AMP-activated S/T protein kinase (*prk*-10), and a transcription factor (*cre*-1), showed no sorbose resistance. Double mutant analysis revealed that the deletion of *hxk*-2, *prk*-10, and *cre*-1 did not affect the sorbose resistance of the *col*-26 mutant. Genes for a glucoamylase (*gla*-1), an invertase (*inv*), and glucose transporters (*glt*-1 and *hgt*-1) were highly expressed in the *cre*-1 mutant, even in glucose-rich conditions, but this upregulation was suppressed in the Δ*cre*-1; Δ*col*-26*a* double-deletion mutant. Furthermore, we found that a *dgr*-2(L1)*a* mutant with a single amino-acid substitution, S11L, in the F-box protein *exo*-1 displayed sorbose resistance, unlike the deletion mutants of *exo*-1, suggesting that the function of *exo*-1 is crucial for the resistance. Our data strongly suggest that CCR directly participates in sorbose resistance, and that *col*-26 and *exo*-1 play important roles in regulating the amylase and glucose transporter genes during CCR.

## 1. Introduction

Filamentous fungi have elongated hyphae at their growing tips with hyphal branches, and undergo radial growth on agar medium. Sorbose, a rare sugar, exhibits toxic effects on several fungi, including *Neurospora crassa*. When 1% sorbose is added to medium in the presence of 0.2% sucrose, *N. crassa* propagates hyper-branched hyphae and forms compact colonies [1,2,3,4]. The formation of compact colonies in the presence of sorbose is a useful methodological tool for the isolation of mutants, allowing high-resolution genetic analyses and contributing to the establishment of *N. crassa* as a model organism for molecular genetics and biochemistry. Various studies have been conducted to identify the way in which sorbose induces such compact colonies; however, the underlying mechanisms remain largely unclear. Sorbose induces changes in the polysaccharide composition of the cell wall, such as a marked decrease in beta-1,3-glucan [3,4], possibly because a beta-1,3-glucan synthetase is inhibited by the sugar [5]. In contrast, a beta-1,3-glucan synthetase inhibitor, micafungin, and a GPI-anchor biosynthesis inhibitor, aminopyrifen, induced abnormal morphology on a sorbose medium in *N. crassa* mutant strains in the cell wall integrity MAP kinase genes *mak*-1 and *mak*-2, and chitin synthetase genes *chs*-5 and *chs*-7 [6,7]. These observations suggest that sorbose may disturb the synthesis of the fungal cell wall.

Several sorbose-resistant mutants have been isolated and characterized in fungi. *Aspergillus nidulans sorA* mutants show cross-resistance to a glucose analog, 2-deoxyglucose (2-DG), and are defective in sugar uptake [8]. The *sorA* gene has been shown to encode the high-affinity glucose transporter protein MstC [9,10]. In contrast, *sorB* mutation in *A. nidulans* did not confer 2-DG resistance, but instead led to a considerable reduction in phosphoglucomutase activity [8]. In *N. crassa*, six sorbose-resistant mutants (*sor*-1*, sor*-2*, sor*-3, *sor*-4*, sor*-5, and *sor*-6) have been isolated and characterized [11,12,13]. Among them, only the function of the *sor*-4 gene has been identified: it encodes an ortholog of Snf3 that acts as a low-affinity glucose sensor in *Saccharomyces cerevisiae*. In contrast, two other *N. crassa* mutations—*dgr*-3, which confers 2-DG resistance, and *rco*-3, which encodes a regulator for conidiation—are localized to *sor*-4. Therefore, the mutation is referred to as *sor*-4/DGR-3/RCO-3 [14,15]. In *S. cerevisiae*, the function of Snf3 is coupled with that of another glucose sensor, Rgt2, which transduces the glucose signal to Rgt1, the transcription regulator of several glucose transporters (Hxt) for adjusting *HXT* gene expression [16,17]. Furthermore, Snf3 and Rgt2 regulate the expression of carbon catabolite repression (CCR) genes [18]. Mig1, a Cys_2_–His_2_ type zinc-finger transcription factor, acts as a central regulator for CCR gene expression. In glucose-rich conditions, Mig1 is dephosphorylated by the Glc7 phosphatase complex and locates in the nucleus to repress Mig1-dependent genes [19]. A hexose kinase, Hxk2, interacts with Mig1 and regulates the phosphorylation status of Mig1 [20]. When cells are in glucose-depleted conditions, Mig1 is phosphorylated by an AMP-activated protein kinase Snf1, which relocates from the nucleus to the cytoplasm [21]; then, Mig1-dependent genes are derepressed and then expressed to adapt glucose starvation.

Although most fungi use glucose as a primary carbon source, some possess various deconstructing enzymes that hydrolyze polysaccharide for alternative carbon sources such as starch, cellulose, and hemicellulose [18,22,23]. Genes encoding these enzymes are generally regulated by the glucose concentration in their environment and are induced to utilize alternative carbon compounds when cells are exposed to glucose-starved conditions. It is well known that Mig1 ortholog transcription factors also act as a central regulator in filamentous fungi [18]. In *Aspergillus sp.*, CreA, the ortholog of Mig1, controls gene expression associated with the utilization of alternative carbon compounds such as starch, arabinose, and xylose [24,25,26]. CreA from *A. nidulans* or *cre*-1 from *Trichoderma reesei* bind to the promoters of target genes to repress their expression via the consensus motif 5′-SYGGRG-3′ [27,28]. In *N. crassa*, the Mig1 ortholog *cre*-1 acts as a central regulator, and the loss of its function leads to the constitutive expression of cellulase and amylase genes, even under glucose nondepleting conditions [29,30]. The expression of polysaccharide-hydrolyzing enzymes such as amylase, cellulase, and hemi-cellulase in *Aspergillus* species is known to be regulated not only by CreA, but also by another transcriptional regulator, AmyR, a fungal-specific Zn(II)Cys_6_-type transcription factor [26,31]. The deletion mutant of *amyR* in *A. oryzae* exhibited growth defects on starch medium and decreased expression levels of alpha-glucosidases [31]. In *T. reesei*, the mutation of *bglR*, an ortholog of *amyR*, caused growth defects on maltose and starch media and the downregulation of a beta-glucosidase gene [32]. *col*-26, the ortholog of AmyR in *N. crassa*, is considered to play a crucial rule in carbon metabolism in the fungus [33,34]. The deletion mutant of *col*-26 presented a growth defect on various carbon sources, including glucose, fructose, and maltose. *col*-26 has been shown to be involved in the regulation of the expression of amylolytic and cellulolytic enzyme genes. Recently, Li et al. [35] reported that the transcription factor *col*-26 and the glucose sensor *sor*-4/*DGR*-3/*RCO*-3 regulated a low-affinity glucose transporter, *glt*-1, along with two high-affinity glucose transporters, *hgt*-1 and *hgt*-2 [35]. In *N. crassa*, mutants for a probable transcription factor *exoamylase*-1 (*exo*-1) were isolated as hypersecretion strains of alpha-amylase, glucoamylase, invertase, pectinase, and trehalase. Recently, Gabriel et al. [36] revealed that *exo*-1 encodes an F-box protein, and that *exo*-1 is identical to the 2-DG resistance gene *dgr*-2. Filamentous fungi tend to have many F-box proteins (39 F-box protein in *N. crassa*), but their function largely remains unclear. The *frp*-1 gene, an ortholog of *exo*-1, was indispensable for the pathogenicity of a plant pathogenic fungus *Fusarium oxysporum*. This result suggests that a SCF (Skp-Cullin-F-box) ubiquitin ligase complex is involved in the emergence of pathogenicity [37]. In *N. crassa*, high secretion of amylases and invertase in Δ*exo*-1 was dependent on the transcriptional regulator *col*-26 but not *cre*-1 [36].

## 2. Materials and Methods

### 2.1. Fungal Strain and Growth Medium

The *N. crassa* strains used in this study are listed in Table 1. The gene deletion strains generated by the Neurospora Genome Project were obtained from the Fungal Genetics Stock Center (FGSC) [38]. As *cre*-1 *het* strain (FGSC#18633) from KO library was heterokaryon, homo Δ*cre*-1*a* strains were obtained from crossing with wild-type strains. The Δ*col*-26*a* strain was crossed with Δ*hxk*-2*A* and Δ*prk*-10*A* to produce Δ*col*-26;Δ*hxk*-2*A* and Δ*col*-26;Δ*prk*-10*A*, respectively. The Δ*col*-26; Δ*cre*-1*a* was obtained by crossing Δ*col*-26*A* with Δ*cre*-1*a.* In the same way, the Δ*sor*-4*a* strain was used to isolate the Δ*sor*-4;Δ*hxk*-2*a* and Δ*sor*-4; Δ*prk*-10*a* strains. The strains were grown on Vogel’s minimal agar medium containing 1.2% sucrose medium (Vm medium) for filamentous growth, and medium containing 1.0% sorbose and 0.2% sucrose (*sor* medium) for colonial growth at 25 °C [39]. Sorbose resistance was evaluated by filamentous growth on *sor* medium 4 days after conidia inoculation.

### 2.2. Sequencing Analysis of dgr Mutants and Confirming of Gene Replacement of Double Deletion Mutants

Genomic DNAs of the wild-type strain, *dgr* mutants, and double deletion mutants were isolated as described previously [7]. *col*-26 and *hxk*-2 genes in *dgr*-1 or *dgr*-4 mutants were amplified by PCR using primers, col-26_seq_F1 and col-26_seq_R4 for *col*-26 gene, and hxk-2_seq_F1 and hxk-2_seq_R4 for *hxk*-2 gene (Appendix A). PCR products were purified using the QiAquick PCR Purification kit (Qiagen, Tokyo, Japan) and sequenced using sequencing primers listed in Appendix A. For confirming of double mutants, Δ*col*-26;Δ*hxk*-2*A*, Δ*col*-26; Δ*prk*-10*A*, Δ*col*-26;Δ*cre*-1*a*, Δ*sor*-4;Δ*hxk*-2*a*, and Δ*sor*-4;Δ*prk*-10*a* were used; deletion of each gene was confirmed by PCR analysis using primers sets shown in Appendix A (Appendix A).

### 2.3. Sensitivity to 2-Deoxyglucose and Mitochondrial Respiration Inhibitors

A glucose analog, 2-deoxyglucose (2-DG), and two mitochondrial complex III inhibitors, azoxystrobin and antimycin A, were purchased from FUJIFILM Wako Pure Chemical Corporation (Tokyo, Japan). To measure the sensitivity to 2-DG, mycelial disks that were precultured on Vogel’s minimal agar medium containing 1.0% fructose were transferred onto 1% fructose Vogel medium containing 2-DG (0.1% to 0.4%), and fungal growth was assessed after 18 h of cultivation at 25 °C. Azoxystrobin and antimycin A act as specific inhibitors of the mitochondrial respiratory chain by binding to the Qo and Qi sites of the cytochrome bc1 complex, respectively. To determine the sensitivity to azoxystrobin and antimycin A, a series of conidia suspensions (1 × 10^7^ to 1 × 10^3^ cells/mL) was spotted on *sor* medium containing azoxystrobin (0.4 mg/L) and antimycin A (0.4 mg/L), and colony formation was photographed after 2 or 5 days of cultivation at 25 °C.

### 2.4. Gene Expression Analysis by qPCR

Total RNA was isolated as previously described by Noguchi et al. [40]. The conidia (1.0 × 10^6^ cells/mL) were inoculated and precultured in Vogel’s liquid medium containing 1.2% glucose for 22 h at 25 °C on a reciprocating shaker (135 rpm) and then, the growing hyphae were transferred to fresh Vogel’s medium containing glucose or maltose and cultured for 2 h at 25 °C. The resultant mycelia were harvested by filtration using an aspirator, and frozen in liquid nitrogen until RNA isolation. Total RNA was isolated from each sample using a FastRNA Pro Red kit (MP Biomedicals, Tokyo, Japan). Each RNA sample (1 µg of total RNA) was subjected to cDNA synthesis and quantitative real-time reverse transcription PCR (RT-qPCR) using the LightCycler system (Roche Diagnostics, Tokyo, Japan), as described by Yamashita et al. [41]. The mRNA expression of the target genes was quantified using Universal ProbeLibrary (Roche Diagnostics, Tokyo, Japan). Primers and probes used in this study are summarized in Appendix A. The relative gene expression value was calculated by comparing the threshold cycle (Cp), with actin used as the reference gene in the RT-qPCR analysis. A minimum of three biological replicates were performed for each experiment.

### 2.5. Glucose and Sorbose Consumption of Sorbose-Resistant Strains

To measure glucose or sorbose consumption during liquid culture, the conidia (10^6^ cells/mL) were inoculated in Vogel’s minimal liquid medium with 1% glucose or 1% sorbose + 0.2% glucose, and incubated at 25 °C. The culture filtrates were harvested after 18, 24, and 48 h. Glucose concentration in the medium was measured using the F-kit for glucose (J.K. International, Tokyo, Japan) in accordance with the manufacturer’s protocol. The sorbose concentration was measured using the Si–Mo method as described by Katano et al. [42]. To measure the sorbose concentration, 200 μL of culture filtrate was mixed with 800 μL of reaction buffer (6 mL of 1 M Na_2_MoO_4_ and 2 mL of 0.25 M Na_2_SiO_3_ were mixed and adjusted to pH 4.5 using 5 M HCl) and incubated 70 °C for 30 min. The absorbance of the resultant samples was measured at 750 nm. Sorbose concentration in the cultures were calculated from the standard curves. Three biological replicates were performed.

### 2.6. SDS-PAGE Analysis of Extracellular Proteins

Conidia (10^6^ cells/mL) were cultured in Vogel’s minimal medium containing 2% glucose for 5 days on a reciprocating shaker (135 rpm) at 25 °C. The extracellular protein in culture filtrates was concentrated using a Centrifugal Ultrafiltration Filter Unit 3000 (Amicnon Tokyo, Japan), SDS-PAGE was performed with a 8–16% Mini-PROTEIN TGX Gels (BIO-RAD, Tokyo, Japan), and the gel was stained with Coomassie Brilliant Blue.

## 3. Results

### 3.1. Loss of Function of Transcription Factor col-26/dgr-1 Confers Sorbose Resistance in N. Crassa

During fungicide sensitivity screening of the deletion mutants in the transcription factor library of *N. crassa*, we found that the *col*-26 deletion mutant was hypersensitive to azoxystrobin and antimycin A, which are both mitochondrial respiratory chain complex III inhibitors (Appendix A). In *N. crassa*, only the *sor*-4 gene (alternate names of *rco*-3 and *dgr*-3), which encodes a glucose sensor protein, has been identified among sorbose resistance genes. Further characterization of the *col*-26 mutants revealed that the Δ*col*-26*a* strain displayed sorbose resistance, similar to *sor*-4 mutants (Figure 1A). Suspensions of conidia (approximately 10^6^ cells/mL) were spotted on *sor* medium containing 1% sorbose and 0.2% sucrose; the Δ*sor*-4*a* and Δ*col*-26*a* strains, but not the wild-type strain, followed a normal growth pattern. We also found the sorbose resistance of a 2-deoxyglucose (2-DG)-resistant strain *dgr*-1(BE52)*A* (FGSC#4326). The *dgr*-1 mutation has been identified as a frame shift at codon 335 of the *col*-26 protein (Appendix A). These observations are consistent with the conclusion that the loss of function of the transcription factor *col*-26 confers sorbose resistance in *N. crassa*.

To compare the growth rates of Δ*sor*-4*a* and Δ*col*-26*a* strains on Vm medium and *sor* medium, mycelial disks precultured on each medium were transferred and cultured on fresh medium of each type. Both sorbose-resistant strains grew at similar rates to that of the wild-type strain (2.04 mm to 2.16 mm/h) on Vm medium (Figure 2A). On *sor* medium, the growth rate of the *sor*-4(DS(r))*A* strain (1.73 ± 0.13 mm/h) was very similar to that of the *sor*-4(DS(r))*A* strain (1.69 ± 0.18 mm/h) (Figure 2B). There was no statically significant difference in growth rate between the Δ*col*-26a strain and the *sor*-4(DS(r))*A* strain. These two precultured sorbose-resistant strains began linear growth on fresh medium without delay, whereas the wild-type strain did not show linear growth on this medium (Figure 2B). When conidia of these strains were inoculated on Vm medium, filamentous growth of the Δ*sor*-4*a* and Δ*col*-26*a* strains, as well as the wild-type strain, were clearly detectable after 18 h. However, on *sor* medium, all strains formed compact colonies 48 h after inoculation, and only the Δ*sor*-4*a* and Δ*col*-26*a* strains began to grow at the edge of the colonies. These observations suggest that there was a transition stage between colonial growth and linear hyphal growth in the Δ*sor*-4*a* and Δ*col*-26*a* strains on *sor* medium. In addition, conidia of the Δ*sor*-4*a* and Δ*col*-26*a* strains, as well as the wild-type strain, did not germinate on Vm medium containing sorbose as the sole carbon source; therefore, the addition of a normal carbon source, such as sucrose or glucose, is essential for filamentous growth on medium containing 1% sorbose.

### 3.2. Sorbose Resistance, 2-DG Resistance, and QoI Sensitivity of CCR Mutants

The sorbose-resistance factors *col*-26 and *sor*-4 are known to be involved in CCR and their mutant strains are less sensitive to 2-DG, an analog of glucose that cannot be isomerized to fructose; therefore, it is not further metabolized and is often used to select for CCR factors in filamentous fungi [13,43]. The involvement of CCR factors in sorbose resistance prompted us to examine whether the deletion of other CCR factors was involved in sorbose resistance and/or 2-DG resistance. We examined three factors, namely a hexokinase, HXK-2, a major CCR-transcription factor, *cre*-1, and an AMP-activated S/T protein kinase, PRK-10. None of these *N*. *crassa* deletion mutants displayed sorbose resistance (Figure 1B); however, the Δ*hxk*-2*a* mutant was as resistant to 2-DG as the Δ*sor*-4*a* and Δ*col*-26*a* strains (Figure 3A). In contrast, the Δ*prk*-10*a* mutant was hypersensitive to 2-DG. The Δ*cre*-1 mutant was previously reported to have 2-DG resistance when Avicel was used as a carbon source [33], but our results showed that the Δ*cre*-1*a* mutant was somewhat sensitive to 2-DG on agar medium containing fructose as a carbon source. We sequenced the *hxk*-2 gene of the *dgr*-4(KHY7)*a* mutant (FGSC#8287) and found the insertion of a 139 bp fragment within the third exon of *hxk*-2 (Appendix A). This insertion resulted in a frame shift at codon 289 and immature termination at codon 292 in *hxk*-2. This is the first report to show the *dgr*-4 gene is identical to the *hxk*-2 gene.

As described above, Δ*col*-26*a* showed hyper-sensitivity to antimycin A and azoxystrobin, which inhibit mitochondrial complex III by binding the Qi-site and the Qo-site of cytochrome *b*, respectively. Both the Δ*sor*-4*a* and Δ*col*-26*a* mutants exhibited very similar sensitivity to these respiration inhibitors (Figure 4A). Not only sorbose-resistant mutants, but also Δ*hxk*-2*a* and Δ*prk*-10*a* mutants were hypersensitive to antimycin A and azoxystrobin (Figure 4B). In contrast, QoI- and QoI sensitivity in the Δ*cre*-1*a* mutant was almost same to that of the wild-type strain. It is well known that the alternative oxidase AOD-1 reduces sensitivity to complex III inhibitors. Therefore, we analyzed *aod*-1 expression in the Δ*col*-26*a* mutants. The *aod*-1 gene was upregulated by azoxystrobin to the same level as in the wild-type strain, suggesting that QoI sensitivities of CCR mutants are independent with alternative oxidase activity.

### 3.3. Comparison of Sorbose and 2-DG Resistance and Gene Expressions of Double Mutants Isolated from Sorbose-Resistant Mutants and CCR Mutants

To investigate the genetic relationship between glucose repression and sorbose resistance, we crossed the *col*-26 or *sor*-4 mutant with CCR mutants to obtain corresponding double mutants (Appendix A). All isolated double mutants, Δ*col*-26; Δ*cre*-1*a*, Δ*col*-26; Δ*hxk*-2*A*, Δ*col*-26; Δ*prk*-10*A*, Δ*sor*-4; Δ*hxk*-2*a*, and Δ*sor*-4; Δ*prk*-10*a* displayed sorbose resistance similar to the Δ*col*-26*a* and Δ*sor*-4*a* mutants on *sor* medium (Figure 1B). In contrast, similar to the Δ*prk*-10*a* mutant, the Δ*col*-26; Δ*prk*-10*A* and Δ*sor*-4; Δ*prk*-10*a* double mutants were hypersensitive to 2-DG, whereas the Δ*col*-26; Δ*hxk*-2*A*, Δ*sor*-4; Δ*hxk*-2*a*, and Δ*col*-26; Δ*cre*-1*a* mutants were resistant to 2-DG (Figure 3B).

*col*-26 and *cre*-1 have been shown to regulate several carbohydrate-related enzymes involved in CCR. We compared the expression pattern of relevant genes in the Δ*cre*-1*a*, Δ*col*-26*a*, and Δ*col*-26; Δ*cre*-1*a* mutants. First, we selected six genes, *gla*-1 (glucoamylase), *inv* (invertase), *gh*31-3 (alpha-glucosidase), *gh*13-2 (alpha-amylase), *glt*-1 (low-affinity glucose transporter), and *hgt*-1 (high-affinity glucose transporter), and compared their expression in medium with glucose or maltose as the carbon source. These genes, except for *gh*13-2 and *glt*-1, were highly induced in the wild-type strain grown on maltose medium (Figure 5A). As previously reported [29], the deletion of the negative regulator *cre*-1 resulted in high constitutive expression of CCR-related genes (Figure 5B). The inductions of *gla*-1, *inv*, *gh*31-3, *gh*13-2, and *hgt*-1 in the Δ*cre*-1*a* mutant was quite evident in glucose-rich conditions, but minimal in glucose-depleted conditions. In contrast, the expression profiles of all six genes were almost the same in the Δ*col*-26*a* and Δ*sor*-4*a* mutants in both conditions. The expression of *gla*-1, *inv*, *gh*31-3, and *hgt*-1 was slightly upregulated in both the Δ*col*-26*a* and Δ*sor*-4*a* mutants in glucose-rich condition, whereas the expression of *gh*13-2 and *glt*-1 was significantly downregulated in glucose-rich and glucose-depleted conditions. It should be noted that the expression pattern of the Δ*col*-26;Δ*cre*-1*a* double mutant was almost the same as that of the Δ*col*-26*a* and Δ*sor*-4*a* mutants (Figure 5B). These data indicated that derepression by the loss of *cre*-1 function might be overcome by the deletion of the *col*-26 gene.

Several glucose transporters have been reported to be downregulated by the Δ*col*-26*a* and Δ*sor*-4*a* mutants [35]; therefore, we measured the concentrations of extracellular glucose and sorbose during cultivation (Figure 6). The wild-type strain consumed the most glucose after incubation for 24 h when the initial glucose concentration was 1%, meanwhile, more than 60% of glucose remained in the culture medium in the cases of the Δ*col*-26*a* and Δ*sor*-4*a* mutants, suggesting their low uptake of glucose (Figure 6A). In contrast, after 24 h incubation, approximately 90% of sorbose remained unincorporated in the case of the Δ*col*-26*a* and Δ*sor*-4*a* mutants, as well as the wild-type strain (Figure 6B). These data suggest that these sorbose-resistance mutants do not incorporate sorbose more than the wild type and do not assimilate it.

### 3.4. A Single S11L Mutation, but Not the Loss-of-function Mutation of F-box Protein exo-1, Confers Sorbose, and 2-DG Resistances in N. Crassa

A 2-DG-resistant strain in *dgr*-2, *dgr*-2(L1)*a*, has sorbose resistance (Figure 7A). Recently, Gabriel et al. [36] revealed that the *exo*-1 gene encoded a F-box protein, and identified a S11L missense mutation within the *exo*-1 gene in *dgr*-2(L1)*a* strain (Appendix A) [36]. In *N. crassa*, the *exo*-1 mutant produced the maximum extracellular glucoamylase activity in starch-supplemented medium [44]; however, the *exo*-1 deletion mutant did not show any sorbose resistance (Figure 7A), indicating that the single amino-acid substitution in *exo*-1/DGR-2, S11L, confers sorbose resistance as well as 2-DG resistance. As shown in Figure 7B, the *dgr*-2(L1)*a* strain was resistant to 2-DG but Δ*exo*-1*a* was somewhat sensitive to the sugar. Furthermore, we confirmed that the Δ*exo*-1*a* strain displayed hypersecretion of proteins, as described previously [36] (Figure 7C), even though the *dgr*-2(L1)*a* strain did not. In glucose-rich conditions, all four genes—namely *gla*-1, *inv*, *glt*-1, and *hgt*-1—were slightly upregulated in the Δ*exo*-1*a* strain, whereas the expression pattern of the *exo*-1^S11L^ mutant resembles to that of the Δ*col*-26*a* strain (Figure 7D). These results suggest that the phenotypes of Δ*exo*-1*a* differ from those of the *exo*-1^S11L^ strain.

## 4. Discussion

L-Sorbose induces hyperbranching of hyphae and results in colonial growth on agar media in *Neurospora crassa*. Among the six genes identified as conferring sorbose resistance (*sor*-1 to *sor*-6), only *sor*-4, which encodes a glucose sensor protein, has been thoroughly investigated [14,15]. In this work, we revealed two more genes, *col*-26, which encodes a transcription factor, and *exo*-1, which encodes a F-box protein, that are likely involved in sorbose resistance (Figure 1A and Figure 7A).

It is well known that *sor*-4 and *col*-26 are factors with important roles in CCR in *N. crassa*. This prompted us to investigate whether any gene mutants in CCR displayed sorbose resistance. *cre*-1, a homolog of *S. cerevisiae* Mig1 and *A. nidulans* CreA transcription factors, controls glucose repression along with a glucose sensor *sor*-4 and a hexokinase HXK-2 [18]. Similar to homologs in other fungi, *N. crassa cre*-1 acts as a negative regulator of regulons for the utilization of carbohydrates other than glucose; therefore, the deletion of the *cre*-1 gene results in the increased secretion of cellulases, amylases, and beta-galactosidases even in glucose-rich condition [29]. In contrast, the AMP-activated S/T protein kinase PRK-10, an ortholog of Snf1 of *S. cerevisiae*, probably phosphorylates *cre*-1 and results in the release of *cre*-1-mediated repression. As described in *S. cerevisiae* [43], the deletion of *hxk*-2 and *prk*-10 in *N. crassa* resulted in cells that were resistant and hypersensitive to 2-DG, respectively (Figure 3). However, neither the *cre*-1, *hxk*-2, nor *prk*-10 mutants showed sorbose resistance (Figure 1B). These results suggest that sorbose resistance may correlate with catabolite repression, but independently of *cre*-1, *hxk*-2, and *prk*-10.

As *sor*-4 and *col*-26 strains are resistant to 2-DG, we further characterized 2-DG resistance (*dgr*) mutants to identify the missing factors that might connect the *sor*-4 glucose sensor and the *col*-26 transcription factor [13]. Among the four *dgr* mutants *dgr*-1 to *dgr*-4, *dgr*-3 is allelic to *sor*-4 [15], the *dgr*-1 and *dgr*-2 strains were sorbose-resistant, but the *dgr*-4 strain was sensitive to the chemical. From the mapping data of *dgr*-4 (Appendix A) [13], this gene was shown to be closely linked to *al*-2 (0.6%). Physical mapping of data from the genome sequence database indicated that *hxk*-2 localizes very close to *al*-2. The direct sequencing of *hxk*-2 from the *dgr*-4(KHY7)*a* strain revealed that the *hxk*-2 gene was disrupted by an insertional mutation (Appendix A). We also sequenced *sor*-4 and *col*-26 of the *dgr*-1(BE52)*A* strain and found that the *dgr*-1 gene is allelic to the *col*-26 gene (Appendix A). Mapping data indicated the localization of *dgr*-1 and *col*-26 on edge of the left arm of chromosome V was consistent with the interpretation that *dgr*-1 and *col*-26 are the same gene. In contrast, there was no mutation in *sor*-4 and *col*-26 in the *dgr*-2 mutants, suggesting that it is a new gene that confers sorbose resistance. Recently, Gabriel et al. [36] revealed that *dgr*-2 is allelic to *exo*-1, which encodes an F-box protein, by means of *dgr*-2 and *exo*-1 mutant genomes [36]. We confirmed that the *exo*-1^S11L^ missense mutation in sorbose-resistance progenies by genome sequencing of *dgr*-2(L1)*a* and *dgr*-2(L5)*A*; however, deletion mutants of *dgr*-2/*exo*-1 did not display sorbose-resistance (Figure 7A). We also confirmed that an *exo*-1 deletion mutant strain, Δ*exo*-1*a*, secreted exocellular enzymes in large amounts, but the *dgr*-2(L1)*a* strain with the *exo*-1^S11L^ mutation did not (Figure 7C). Moreover, the *dgr*-2(L1)*a* strain displayed 2-DG resistance, but the *exo*-1 deletion mutant did not show clear 2-DG resistance (Figure 7B). Our results indicate that the phenotypes of the *exo*-1 point mutation strain *dgr*-2(L1)*a* and the deletion mutant strain Δ*exo*-1*a* are quite different in these aspects. A serine residue at amino-acid position 11 of the F-box protein is conserved in other fungi, such as *Magnaporthe grisea*, *Fusarium graminearum*, *Botrytis cinerea*, and *Trichoderma reesei* (Appendix A). Although the function of the N-terminal conserved region (consisting of approximately 20 amino acids) is unclear, the serine at 11 may be phosphorylated and affect the function of F-box domain (amino acids 112–144 in *exo*-1). Although its function is still unknown, the F-box protein *exo*-1 may form SCF complexes and induce ubiquitination of protein(s) targeted for degradation by the 26S proteasome: *col*-26 is a possible candidate target. Phenotypes of *exo*-1*^S11L^* strain resemble *col*-26 deletion mutants but not *exo*-1 deletion mutants regarding sorbose resistance and gene expression (Figure 7). Gabriel et al. [36] reported that the loss of function of *exo*-1 induces glucoside enzymes in *cre*-1 independently [36]. One possible explanation is that the *exo*-1 deletion suppresses *col*-26 degradation and leads to the overexpression of glucosidases. In contrast, the S11L mutation results in the constitutive activation of *exo*-1, hyper-ubiquitination, and the degradation of *col*-26.

As previously reported [29,30], the loss of *cre*-1 function leads to the overexpression of *gla*-1, *gh*13-2, *gh*31-3, *inv*, *glt*-1, and *hgt*-1*,* even in glucose-rich conditions (Figure 5A). *cre*-1-like transcription factors are the main negative regulators in glucose repression in many fungi [18]. The expression of *glt*-1, which encodes a low-affinity glucose transporter, in three sorbose-resistance mutants, namely Δ*col*-26*a*, Δ*sor*-4*a*, and Δ*col*-26;Δ*cre*-1*a* was significantly reduced, even in glucose-rich conditions, suggesting that the glucose sensor *sor*-4 and the transcription factor *col*-26 are essential for the expression of *glt*-1, as described elsewhere [35]. In addition, the expression of *gh*13-2, which encodes an alpha-amylase, in these three mutants was comparable to that of *glt*-1, suggesting that these two genes would be controlled under the same system. We noticed that the gene expression profiles for carbohydrate-related genes in *col*-26 and *sor*-4 were similar. Moreover, a double mutant Δ*col*-26;Δ*cre*-1*a* had a very similar gene expression profile to Δ*col*-26*a* and Δ*sor*-4*a*. It should be noted that the upregulation found in the *cre*-1 strain was mostly negated in the Δ*col*-26;Δ*cre*-1*a* strain. The *col*-26 transcription factor could positively control its target genes, which should be suppressed by unphosphorylated *cre*-1, and the loss cannot be compensated for because of the absence of the negative regulator *cre*-1. Indeed, the Δ*col*-26;Δ*cre*-1*a* strain displayed a more similar phenotype to the *col*-26 single mutant than the *cre*-1 mutant. Meanwhile, the expression of *gla*-1, *inv*, *gh*31-3, and *hgt*-1 in these three sorbose-resistant mutants in glucose-rich conditions was slightly upregulated. The presence of any *cre*-1 and *col*-26 independent regulatory system (s) cannot be eliminated.

The mechanism underlying sorbose resistance in *N*. *crassa* is still unclear. In our study, we revealed that, in addition to the glucose sensor *sor*-4, two factors—namely *col*-26 and *exo*-1—are involved in sorbose resistance. All evidence obtained regarding sorbose resistance indicates the connection of the expression of carbohydrate-related genes in glucose-depleted conditions and the toxicity of the chemical. Sorbose might disturb signaling pathway(s) concerning carbohydrate-related gene expression by interacting with the sugar-sensing system. *col*-26 upregulates glucose transporter genes, including *glt*-1 and *hgt*-1 [35]; indeed, *col*-26 mutant uptake less glucose than the wild-type strain. We speculate that sorbose-resistant mutants hardly assimilate the sugar, as its incorporation by the *col*-26 mutants was only marginally different to the that of the wild-type strain (Figure 6).

CCR is fundamental mechanism using proper energy sources; therefore, it is conserved in prokaryote and eukaryotes. In fungi, CCR is directly connected with various applications, such as the production of Japanese sake and miso by *Aspergillus* species, bioethanol production by *T. reesei*, and plant protection from plant pathogenic fungi. Recent studies indicate that catabolite repression in fungi is connected to several signaling molecules, such as cAMP-dependent protein kinase and the stress-response MAP kinase. Therefore, our results in the model fungus *N. crassa* will contribute to elucidation of the complex mechanism of fungal CCR.

## Figures and Tables

**Figure 1 jof-08-01169-f001:**
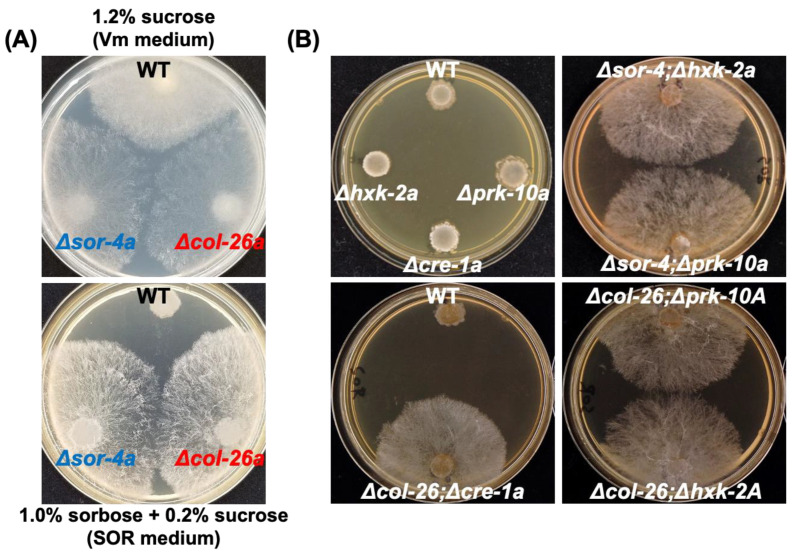
Sorbose resistance of deletion mutants of *sor*-4 and *col*-26 gene, and their double mutant, with the factors involved in carbon catabolite repression. (**A**) Sorbose resistance of *sor*-4 and *col*-26 mutants. Conidia suspensions of wild type, Δ*sor*-4*a*, and Δ*col*-26*a* strains were inoculated and cultured on Vogel’s minimal medium containing 1.2% sucrose (Vm medium, top panel) for 2 days, and on Vogel’s minimal medium containing 1% sorbose and 0.2% sucrose (*sor* medium, bottom panel) for 4 days. Both Δ*sor*-4*a* and Δ*col*-26*a* displayed filamentous growth on *sor* medium. (**B**) Sorbose resistance in mutants of carbon catabolite repression and their double deletion mutants with *col*-26 or *sor*-4 mutant. The deletion mutants of *hxk*-2 (hexokinase), *prk*-10 (AMPK protein kinase), and *cre*-1 (transcriptional repressor) genes did not possess sorbose resistance on *sor* medium, but their double mutants with *sor*-4 or *col*-26 deletion were sorbose-resistant.

**Figure 2 jof-08-01169-f002:**
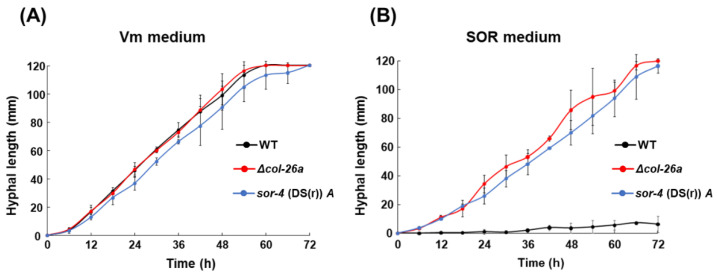
Comparison of growth rates of the *col*-26 and *sor*-4 mutants on Vm medium and *sor* medium. (**A**) Growing hyphae of the wild type, *sor*-4(DS(r))*A*, and Δ*col*-26*a* strains on Vm medium were transferred onto fresh Vm medium. (**B**) Growing hyphae of the *sor*-4(DS(r))*A* and Δ*col*-26*a* strains on *sor* medium were transferred onto fresh *sor* medium. For the wild-type strain, conidia were spread on the *sor* medium and incubated for 2 days, and then agar disks containing growing cells were transferred onto fresh *sor* medium. Hyphal elongation (mm) from the edges of inoculation disks were measured for 3 days. Errors are expressed as standard deviation.

**Figure 3 jof-08-01169-f003:**
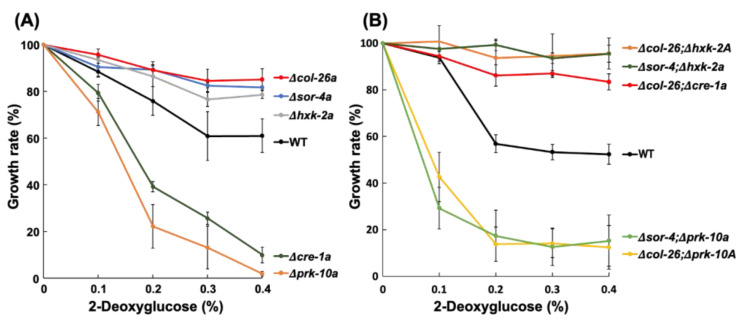
2-Deoxyglucose resistance of the CCR mutants. (**A**) Single deletion mutants of *hxk*-2*, prk*-10*, cre*-1, *col*-26 and *sor*-4. (**B**) Their double mutants of *col*-26; *hxk*-2*, col*-26; *prk*-10*, col*-26; *cre*-1*, sor*-4; *hxk*-2, and *sor*-4; *prk*-10. Mycelium disks precultured on Vogel’s minimal medium containing 1% fructose were inoculated on fructose medium containing 2-deoxyglucose (0.1%, 0.2%, 0.3%, and 0.4%) for 18 h at room temperature. The growth rate was calculated following formula: Growth rate (%) = {hyphal length of treatment (mm)/hyphal length of control (mm)} × 100. Three biological replicates were performed. Errors are expressed as the standard error.

**Figure 4 jof-08-01169-f004:**
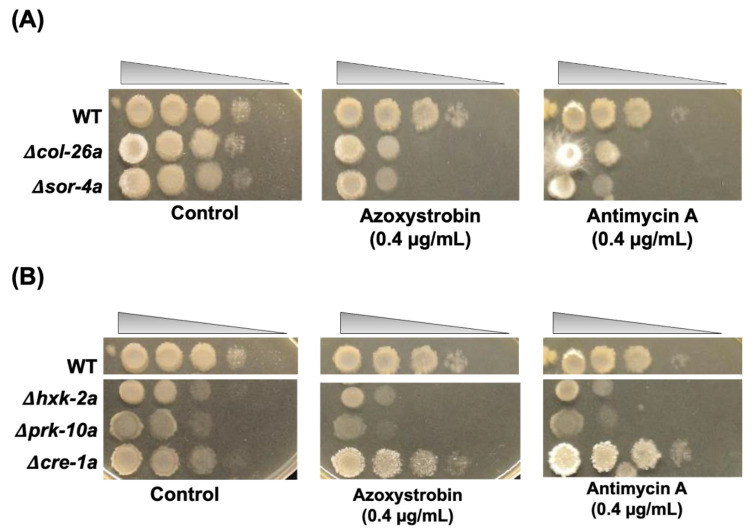
Sensitivity to the mitochondrial complex III inhibitors azoxystrobin and antimycin A. A series of conidia suspensions (10^7^–10^3^ cells/mL) was spotted on *sor* medium containing azoxystrobin (0.4 mg/L) and antimycin A (0.4 mg/L) at room temperature. Control plates and fungicide treatment plates were photographed at 2 days and 5 days after inoculation, respectively. (**A**) *col*-26 and *sor*-4 strains. (**B**) *hxk*-2, *prk*-10 and *cre*-1 strains.

**Figure 5 jof-08-01169-f005:**
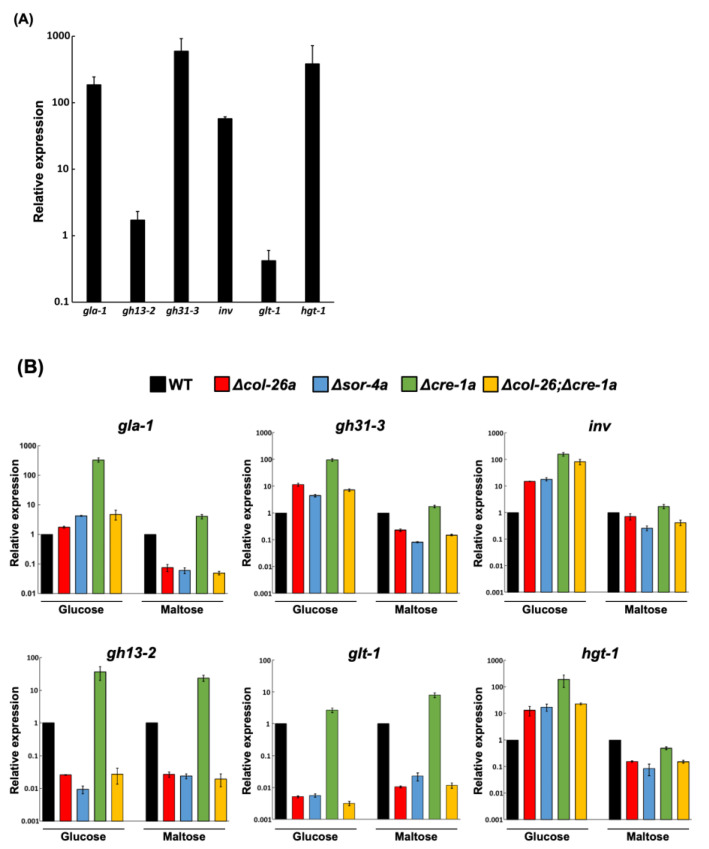
Gene expression analysis of the *col*-26, *sor*-4, *cre*-1 and *cre*-1; *col*-26 mutants. (**A**) Gene upregulation in the wild-type strain under glucose derepression conditions shifted from 1.2% glucose medium to 1.2% maltose medium (see Materials and Methods). Six genes, *gla*-1 (glucoamylase), *gh*13-2 (alpha-amylase), *gh*31-3 (alpha-glucosidase), *inv* (invertase), *glt*-1 (low-affinity glucose transporter), and *hgt*-1 (high-affinity glucose transporter) were selected as target genes for qPCR analysis. With the exception of *gh*13-2 and *glt*-1, all genes were highly upregulated in maltose medium. (**B**) Comparison of gene expression patterns in the Δ*col*-26*a*, Δ*sor*-4*a,* Δ*cre*-1*a* and Δ*col*-26; Δ*cre*-1*a* mutants. Expression levels in each mutant strain were calculated relative to the wild-type strain grown under glucose-rich conditions (Glucose) and glucose derepression conditions (Maltose). Errors are expressed as the standard error. At least three biological replicates of each experiment were performed.

**Figure 6 jof-08-01169-f006:**
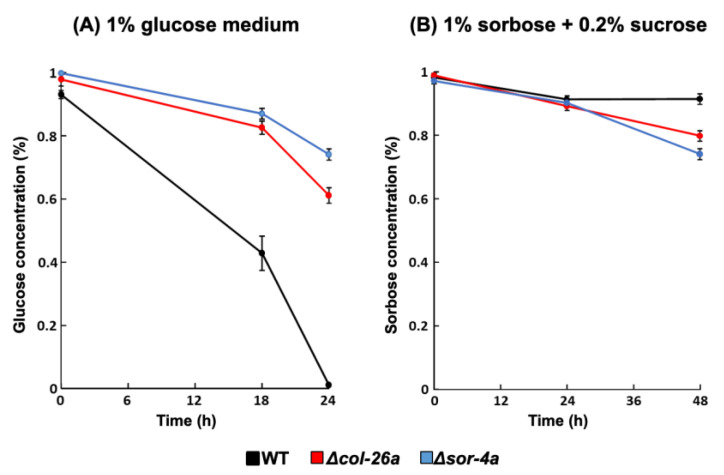
Consumption of glucose or sorbose in sorbose-resistant *sor*-4 and *col*-26 mutants. Glucose (**A**) and sorbose (**B**) concentration in the culture medium in the wild type, Δ*col*-26*a*, and Δ*sor*-4*a* strains. The concentration of glucose and sorbose in culture filtrate was measured using the F-kit for glucose and the Si–Mo method (see Materials and Methods), respectively. Errors are expressed as the standard error. Three biological replicates were performed.

**Figure 7 jof-08-01169-f007:**
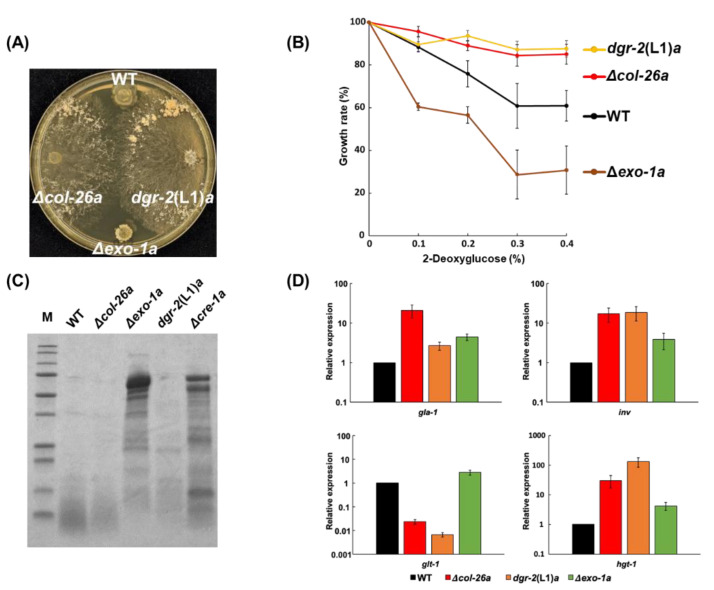
Comparison of the phenotypes of the *exo*-1^S11L^ (*dgr*-2(L1)*a*) and Δ*exo*-1*a* mutants. Sorbose resistance (**A**) and 2-DG resistance (**B**) of wild-type, Δ*col*-26*a*, *exo*-1^S11L^ (*dgr*-2(L1)*a*), and Δ*exo*-1*a* strains. (**C**) SDS-PAGE analysis of proteins secreted by each strain. The extracellular protein in culture filtrates was concentrated and applied for SDS-PAGE analysis. (**D**) Gene expression of *gla*-1, *inv*, *glt*-1, and *hgt*-1 under glucose-rich conditions (see Figure 5) in each strain. Errors are expressed as the standard error.

**Table 1 jof-08-01169-t001:** Strain list used in this study.

Strain	Gene Locus	Genotype (Allele)	FGSC No.	Reference
wild type		*mat a* (ORS-SL6)	4200	FGSC ^a^
Δ*col*-26*a*	NCU07788	*mat a*; *col*-26::*Hyg ^r^*	11030	FGSC ^a^
Δ*col*-26*A*	NCU07788	*mat A*; *col*-26::*Hyg ^r^*	11031	FGSC ^a^
Δ*sor*-4*a*	NCU02582	*mat a*; *sor*-4::*Hyg ^r^*	17928	FGSC ^a^
*sor*-4(DS(r))*A*	NCU02582	*mat A*; *sor*-4(DS(r))	1741	FGSC ^a^
Δ*hxk*-2*a*	NCU00575	*mat a*; *hxk*-2::*Hyg ^r^*	15921	FGSC ^a^
Δ*hxk*-2*A*	NCU00575	*mat A*; *hxk*-2::*Hygr*	15920	FGSC ^a^
Δ*prk*-10*a*	NCU04566	*mat a*; *prk*-10::*Hyg ^r^*	12420	FGSC ^a^
Δ*prk*-10*A*	NCU04566	*mat A*; *prk*-10::*Hyg ^r^*	12421	FGSC ^a^
*cre*-1 *het*	NCU08807	*mat a*; *cre*-1::*Hyg ^r^* (heterokaryon)	18633	FGSC ^a^
Δ*cre*-1*a*	NCU08807	*mat a*; *cre*-1::*Hyg ^r^*		this study ^b^
Δ*exo*-1*a*	NCU09899	*mat a*; *exo*-1::*Hyg ^r^*	19860	FGSC ^a^
*dgr*-1(BE52)*A*		*mat A*; *dgr*-1(BE52)	4326	FGSC ^a^
*dgr*-4(KHY7)*a*		*mat a*; *dgr*-4(KHY7)	8287	FGSC ^a^
*dgr*-2(L1)*a*		*mat a*; *dgr*-2(L1)	4328	FGSC ^a^
*dgr*-2(L5)*A*		*mat A*; *dgr*-2(L5)	4329	FGSC ^a^
Δ*col*-26; Δ*hxk*-2*A*		*mat A*; *col*-26::*Hyg ^r^*; *hxk*-2::*Hyg ^r^*		this study ^b^
Δ*col*-26; Δ*prk*-10*A*		*mat A*; *col*-26::*Hyg ^r^; prk*-10*::Hyg*		this study ^b^
Δ*col*-26; Δ*cre*-1*a*		*mat a*; *col*-26::*Hyg ^r^; cre*-1::*Hyg ^r^*		this study ^b^
Δ*sor*-4; Δ*hxk*-2*a*		*mat a*; *sor*-4::*Hyg ^r^; prk*-10::*Hyg ^r^*		this study ^b^
Δ*sor*-4; Δ*prk*-10*a*		*mat a*; *sor*-4::*Hyg ^r^*; *hxk*-2::*Hyg ^r^*		this study ^b^

^a^ Neurospora mutant strains were obtained from Fungal Genetics Stock Center. ^b^ Strains were obtained by cross (see Materials and Methods). *Hyg ^r^* is hygromycin resistance gene.

## Data Availability

All data are available in the main text or the Appendix A.

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
