# Peer review of "Deletion of the col-26 Transcription Factor Gene and a Point Mutation in the exo-1 F-Box Protein Gene Confer Sorbose Resistance in Neurospora crassa"

_jof, 2022, doi:10.3390/jof8111169_

Round 1

Reviewer 1 Report

The manuscript is interesting manuscript and the work seems to be appropriate with respect to the conclusions. This paper presents a detailed study of the implication of COL-26 and EXO-1 being involved in sorbose resistance. These contribute to the elucidation of the complex mechanism of fungal CCR. Several problems need to be added and considered by the authors:

1. Since the expression levels of glucose transporter genes (glt-1, hgt-1) were inconsistent in the Δcol-26a mutants, how to explain the consumption of glucose or sorbose in sorbose-resistant sor-4 and col-26 mutants in Fig.6?

2 EXO-1 is found to be involved in sorbose resistance, I would suggest introducing the EXO-1 in fungi in the section of the introduction.

3. How to explain the mutant with S11L in the EXO-1 displayed sorbose resistance, while the deletion mutants of exo-1 did not.

4. Is there a potential relationship between COL-26 and CRE-1, since CRE-1 is a central regulator for CCR in filamentous fungi?

5. What do the experiments on protein secretion in Result 3.4 (Fig. 7B) illustrate?

6. Figure legend about “hgt-2 (high-affinity glucose transporter)” (line 279) does not seem to be mentioned in the manuscript and figure.

7. The order of letters in Fig. 7 does not correspond to the figure (line 317-320), please check.

Reviewer 2 Report

The experiments were well designed and the findings are important to the peers. However, the language needs thorough and careful revision. In addition, the authors also need to address the following questions. 

1. Title: The title should be straightforward, wrapup, and directly describes the function of col-26 and exo-1 F-box in CCR, instead of narrating the results of the mutants here.

2. Author name: "Kenshi, Hirai", remove the comma.

3. Affiliation: "Faculty of ..." should be corrected, only the affiliations are needed, "Faculty of." should be deleted.

4. Abstract:

1) "L-Sorbose induces hyperbranching of hyphae and results", which results in.

2) The sentence in lines 11-12, please rephrase it. It can be expressed as "XXXX genes have been confirmed in XXXXXXX".

3) The sentence in lines 13-14 needs rewriting, it has three "that".

4) before the sentence of "The deletionmutants ...", "in contrast" or similar expression could be added to reflect the logic association. "Did not display" could be rewritten as "showed no ...".

5) lines 18-19: highly? Should be "higher" compared to something? other strains?

6) Line 22: is crucial.

7) Line 23: "participates directly" should be directly participates.

8) Line 24: "in gene regulation during CCR", in regulating .... Here the authors should specify how these gene regulate? What are the effects or features?

5. Introduction:

1) Lines 33-34:"the mechanisms underlying this", the underlying mechanisms.

2) Lines 50-52: rephrase this long sentence.

3) Line 54: "transcriptional regulator", transcription regulator 

6. Results:

1) Paragraph from lines 159-173: "Neurospora crassa", here the authors can use the abbreviation.

2) Fig. 1 legend: in the (A) panel, "top panel" and "bottom panel" can be added as detail. In (B), were these plates prepared in the same time? Otherwise, WT needs to be innoculated on every plates.

3) Fig. 2: Please add error bars and indicate the differences with statistical significance.

4) Line 229: The subtitle should be a phrase like others.

5) Fig. 3: n=?

6) Lines 247-248: The subtitle is too long

7) Fig. 5: "The error bar shows the standard error", please write such information by following similar papers in this journal, which applies to all the figures in this manuscript.

8) "of exo-1", remove additional space.

7. Supplemental materials

1) Fig. S1, please adjust locations of the red rectangles to exactly indicate the colonies of interest but not the neighbouring colonies.

2) The resolution of Fig. S4B is low.

Reviewer 3 Report

The manuscript “Deletion of the col-26 transcription factor gene and a point mutation in the exo-1 F-box protein gene confer sorbose resistance in Neurospora crassa” by Hirai et al. reports deletion mutants of col-26, a gene that encodes an AmyR-like transcription factor that also participates in CCR, displayed sorbose resistance. This work is well designed and written. Therefore, I would suggest authors may take a minor revision before publication. Here are the comments and suggestions:

1.     If there any model related with the results in Fig. 2?

2.     Some mutations in Fig. 3(A) seem not stable around 0.2-0.4 % of 2-Deoxyglucose, did authors have any explanation on this?

3.     Authors may plot Figs. 3 and 7(C) with some models, which may relate with the effect of 2-Deoxyglucose resistance.

Round 2

Reviewer 2 Report

Lines 6,7: The affiliations seem wrong as there are no terms of school or department or center.

Line 90: It should be "remains largely unclear"

Line 89: There is something wrong with the space between "protein" and "the". Also in line 11.

Lines 91-93: This sentence needs rephrasing.

Line 35: it should be "underlying".

Line 23: remove "and".

Lines 200: use singular and change it to be "either ... or ..."

Figure 4 is confusing: at the top of each panel there is a slope which I suppose to represent the concentration gradient, however, there is concentratioin labeled at the bottom. The authors may label the figures of cell numbers at the two ends of the slopes, and add labels like "Cell density".

Figure 5A: Which one is the control? Were the levels of other genes normalized according to the control? 5Aand 5B: Why the Y-axis uses multiplicator? Is it because the fold change is not significant?

Other: I suggest the authors have the language polished by people who use English as negative language or commercial editing service, since the expression in many places could be improved. 
